# A Self-Calibrated Non-Parametric Time Series Analysis Approach for Assessing Insect Defoliation of Broad-Leaved Deciduous *Nothofagus pumilio* Forests

**Roberto O. Chávez** [1,*] , **Ronald Rocco** [1] , **Álvaro G. Gutiérrez** [2] , **Marcelo Dörner** [3] **and Sergio A. Estay** [4,5]

1 Instituto de Geografía, Lab. Geo-Información y Percepción Remota, Pontificia Universidad Católica de Valparaíso, Valparaíso 2362807, Chile; roberto.chavez@pucv.cl
2 Departamento de Ciencias Ambientales y Recursos Naturales, Universidad de Chile, Santiago 8820808, Chile; bosqueciencia@gmail.com
3 Corporación Nacional Forestal (CONAF), Aysén 8330407, Chile; marcelo.dorner@conaf.cl
4 Instituto de Ciencias Ambientales y Evolutivas, Universidad Austral de Chile, Valdivia 5110566, Chile; sergio.estay@uach.cl
5 Center of Applied Ecology and Sustainability, Pontificia Universidad Católica de Chile, Santiago 8331150, Chile
* Correspondence: roberto.chavez@pucv.cl; Tel.: +56-32-2274090

**Abstract:** Folivorous insects cause some of the most ecologically and economically important disturbances in forests worldwide. For this reason, several approaches have been developed to exploit the temporal richness of available satellite time series data to detect and quantify insect forest defoliation. Current approaches rely on parametric functions to describe the natural annual phenological cycle of the forest, from which anomalies are calculated and used to assess defoliation. Quantification of the natural variability of the annual phenological baseline is limited in parametric approaches, which is critical to evaluating whether an observed anomaly is "true" defoliation or only part of the natural forest variability. We present here a fully self-calibrated, non-parametric approach to reconstruct the annual phenological baseline along with its confidence intervals using the historical frequency of a vegetation index (VI) density, accounting for the natural forest phenological variability. This baseline is used to calculate per pixel (1) a VI anomaly per date and (2) an anomaly probability flag indicating its probability of being a "true" anomaly. Our method can be self-calibrated when applied to deciduous forests, where the winter VI values are used as the leafless reference to calculate the VI loss (%). We tested our approach with dense time series from the MODIS enhanced vegetation index (EVI) to detect and map a massive outbreak of the native *Ormiscodes amphimone* caterpillars which occurred in 2015–2016 in Chilean Patagonia. By applying the anomaly probability band, we filtered out all pixels with a probability <0.9 of being "true" defoliation. Our method enabled a robust spatiotemporal assessment of the *O. amphimone* outbreak, showing severe defoliation (60–80% and >80%) over an area of 15,387 ha of *Nothofagus pumilio* forests in only 40 days (322 ha/day in average) with a total of 17,850 ha by the end of the summer. Our approach is useful for the further study of the apparent increasing frequency of insect outbreaks due to warming trends in Patagonian forests; its generality means it can be applied in deciduous broad-leaved forests elsewhere.

**Keywords:** insect outbreak; npphen; kernel density; pest management; forest monitoring

---

## 1. Introduction

Insect outbreaks are considered one of the major disturbances for temperate forests in North America and Europe, leading to extensive timber and carbon losses [1–3]. These natural events have had dramatic consequences not only for the forestry industry, but also for the ecosystem and biodiversity conservation related to changes in the forest carbon cycle, composition, and structure [4,5]. In particular, folivorous insects remove the photosynthesizing tissue of leaves and reduce carbohydrate production [6] and are considered among the most ecologically and economically significant groups of forest pests worldwide [7–9]. Different from the Northern Hemisphere, where most of the scientific literature documenting insect forest defoliation has been carried out [10], in the Southern Hemisphere there are few reports about insect outbreaks in forests. For example, in temperate forests in the southern tip of South America (Chilean and Argentinian Patagonia), massive insect outbreaks of the native moth *Ormiscodes amphimone* causing total defoliation of broad-leaved *Nothofagus pumilio* (Nothofagaceae) forests have recently been reported [11–13]. However, there is a lack of studies accounting for the spatial dimension of the defoliation level of these outbreaks despite their being in high demand for forest conservation and pest management. A major reason for this research gap is that collecting field data to assess the defoliation level of folivorous insect outbreaks in remote and vast areas such as Chilean Patagonia is challenging.

Remote sensing emerges as a possibility to assess insect outbreaks of inaccessible forests. Defoliation, i.e., a reduction on the leaf area index (LAI) of the forest, has an effect on the light reflection and absorption properties of the vegetation canopy, particularly a decrease in reflection in the near-infrared region, which can be retrieved by multispectral and hyperspectral sensors [14–19]. In most remote sensing-based assessments, image classification is applied to map the damaged area (extent) or simple remote sensing VIs such as the normalized difference vegetation index (NDVI) or the enhanced vegetation index (EVI) are used as a bi-temporal proxy (before and after the outbreak) to quantify the green biomass loss [10,19]. However, these methods need field data to assist with image classification or algorithm calibration. As the availability and temporal framework of remote sensing time series have increased, studies have incorporated the temporal dimension by using dense time series of satellite images such as Landsat (16-day temporal resolution), Moderate Resolution Imaging Spectroradiometer (MODIS) (16-day, 8-day, and 1-day), and SPOT (1-day) [20–24]. Using this more detailed temporal dimension, insect defoliation can be accurately detected and quantified as a deviation from the natural temporal trajectory (leaf phenological cycle) on a "near-real time" basis [25–27].

Different methodologies have been developed or adapted to quantify insect forest defoliation using dense time series of VIs. For example, Eklundh et al. [28] detected insect defoliation in Scots pine using the seasonal profiles of VIs (outbreaks caused negative slopes for linear functions fitted to NDVI values during the peak of the growing season (GS) when positive slopes were expected for a normal year) along with differences in summer mean values. These measures were used to classify areas into "damaged" and "undamaged" forest using a Boolean combination of threshold parameters. Anees and Ayral [27] tested two methods to detect beetle attacks on pine forests: the first method was based on a moving average finite impulse response (FIR) filter and the second based on a cosine model with three modulated parameters (mean, phase, and amplitude) to describe the "undisturbed" situation [29], from which defoliation was assessed using different VIs. Another example is the method proposed by Olsson et al. [26] to assess insect defoliation in birch forests, where double logistic functions were fitted to a MODIS NDVI time series to set the reference period on a pixel basis. After that, the baseline annual curve was calculated as a weighted average per date using the MODIS quality assessment bands to assign weights (higher qualities were assigned higher weights). Once the baseline curve was set, values filtered by outbreak year were contrasted to detect defoliation. Recently, Pasquarella et al. [25] used an approach similar to the Breaks for Additive Seasonal and Trend (BFAST) [30] method to study defoliation caused by gypsy moth in oak and aspen forests. They created "synthetic" Landsat images for the reference period by fitting harmonic models to a time series of 11 years of Landsat data (they applied a physically-based transformation to Landsat data, tasseled cap greenness, which

combines individual Landsat spectral bands into a single vegetated greenness index) and averaging the smoothed values at a pixel level. Observed values were then contrasted to the reference "synthetic" image. A common feature of the all above mentioned methods is that they rely on parametric models (cosine functions [22,27], double logistic functions [26] or harmonic functions [25]) to describe the "undisturbed" condition (prior to the outbreak), and therefore depend closely on the quality of the fitting exercise. Once the parametric curve has been fitted, observations during or after the outbreak are contrasted to this "undisturbed" baseline to account for the occurrence and/or magnitude of insect defoliation. However, and unlike other remote sensing products, no reliability bands are provided with the insect defoliation output rasters.

Besides the lack of a reliability band, another disadvantage of these parametric-based methods is that they assume that the annual phenological cycle of forests can be well explained by a parametric curve assuming a pre-defined behavior (cosine or double logistic function or a curve built up with harmonic functions), which may not be representative of some species, for example in semi-arid ecosystems [31]. Furthermore, the natural interannual variability of the forest phenology is lost when averaging the annual VI for different years to construct the "undisturbed" baseline, making difficult to discriminate whether an observed anomaly is only part of the natural forest phenological variability or is an actual insect defoliation. A quantification of the forest phenological variability using a parametric approach was introduced by Olsson et al. [32]. In this study, and after the VI time series were smoothed using parametric functions, z-scores of the maximum values of the growing season were calculated for the baseline period to define significant anomalies. Nevertheless, this method is limited to only one phenological metric (seasonal maximum) for each GS and does not deliver a reliability band along with the anomaly output raster.

In order to overcome these limitations, we propose a flexible non-parametric approach based on probabilistic estimations of the annual phenological cycle, from which anomalies can be assessed in terms of the frequency distribution of historical records. In other words, an anomaly on a given date can be calculated as the difference between the observed value and the "most expected" value on that date. Then, the observed anomaly can be only considered "true" if, for example, the observed value falls outside the 95% frequency of historical records. We tested our method to study a large spatial scale defoliation event caused by an *O. amphimone* outbreak in *N. pumilio* forests in the Chilean Patagonia. Since *N. pumilio* is a strictly deciduous species [33] and considering the restricted accessibility of most forests in this region to gather ground truth data, we developed a self-calibrated approach using the winter values of our annual phenological reconstruction as a baseline for total defoliation at the pixel level. Thus, the present research introduces a novel non-parametric and self-calibrated approach for assessing *O. amphimone* outbreaks in deciduous *N. pumilio* forests using dense time series of the MODIS Terra-Aqua EVI combined (8-day temporal resolution). Specifically, we aimed at (1) mapping the insect defoliation by means of EVI anomalies from a probabilistic reconstructed annual phenology (2) delivering an anomaly probability band to filter out non-significant EVI anomalies on a pixel basis, (3) assessing the insect defoliation map for one specific date with a field survey of the outbreak event, and (4) studying the impact of insect defoliation on the LAI of the affected *N. pumilio* forests.

## 2. Materials and Methods

### 2.1. Study Area

During the austral summer 2015–2016 an alert from the Chilean Forest Service (CONAF) attracted the attention of the scientific community and authorities to an unprecedented massive insect outbreak affecting *N. pumilio* forests in the Trapananda National Reserve (45° S, 72° W, Figure 1). The shocking eruption of *O. amphimone* caterpillars and the defoliated forest were visible across hundreds of ha within the protected area and beyond, affecting local activities like farming, forestry, and tourism, with tourism being one of the most important economic activities in the Aysén administrative region (Chilean Patagonia). *O. amphimone* is considered detrimental for tree growth and timber production,

can kill saplings, and potentially cause crown dieback on *N. pumilio* if defoliation is severe [34]. Due to its relevance, we considered this case study as an excellent opportunity to test our approach and, at the same time, to provide a thorough assessment of the insect outbreak.

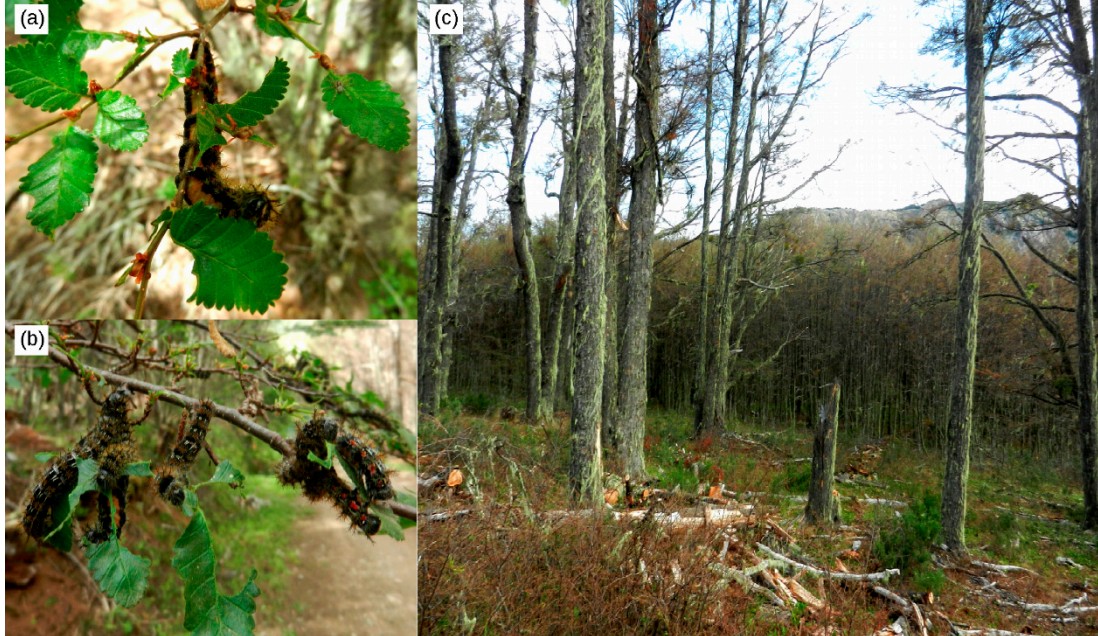

**Figure 1.** *Ormiscodes amphimone* outbreak affecting deciduous broad-leaf *Nothofagus pumilio* forests in the Trapananda National Reserve, southern Chile: (**a**) An individual of *O. amphimone* caterpillar feeding on *N. pumilio* leaves; (**b**) Group of caterpillars defoliating a *N. pumilio* tree; (**c**) A totally defoliated *N. pumilio* forest (grey forest) after the outbreak event of 2015–2016. All pictures were taken by the authors during a field campaign in the Trapananda National Reserve in February 2016. Location of the Reserve is given in Figure 2.

We define the study area as the Mañihuales watershed, which encompasses the Trapananda National Reserve (Figure 2). The watershed comprises 423,026 ha with altitudes ranging from 18 m.a.s.l. downstream in the Mañihuales river to 2042 m.a.s.l. in the mountainous area towards the Andes. The mean altitude of the watershed is 918 m.a.s.l. 47% of the watershed is covered by native forest (198.715 ha), of which 134,525 ha (67%) are broad-leaf forests of *N. pumilio*. Other relevant land covers are bare areas (14% of the watershed area), shrublands (12%), grasslands (5%) and glaciers (3%). The National Reserve has 1369 ha of native lenga forest, which is 77% of its area (Figure 2). Native forests in the Mañihuales watershed include the southernmost distribution of the Magellanic or sub-Antarctic forests [35]. Magellanic forests are composed of evergreen *Nothofagus betuloides* forests in rainy coastal areas and deciduous *N. pumilio* forests in dryer and colder climates [36,37], which is the case of the Mañihuales watershed. Of these, *N. pumilio* is the main host of *O. amphimone*, as was the case of the forested area defoliated in the Trapananda National Reserve during the insect outbreak of the 2015–2016 GS. GSs in the Southern Hemisphere start by convention in July and end in June of the following year, e.g., from 1 July 2015 to 30 June 2016 in the case of the 2015–2016 GS.

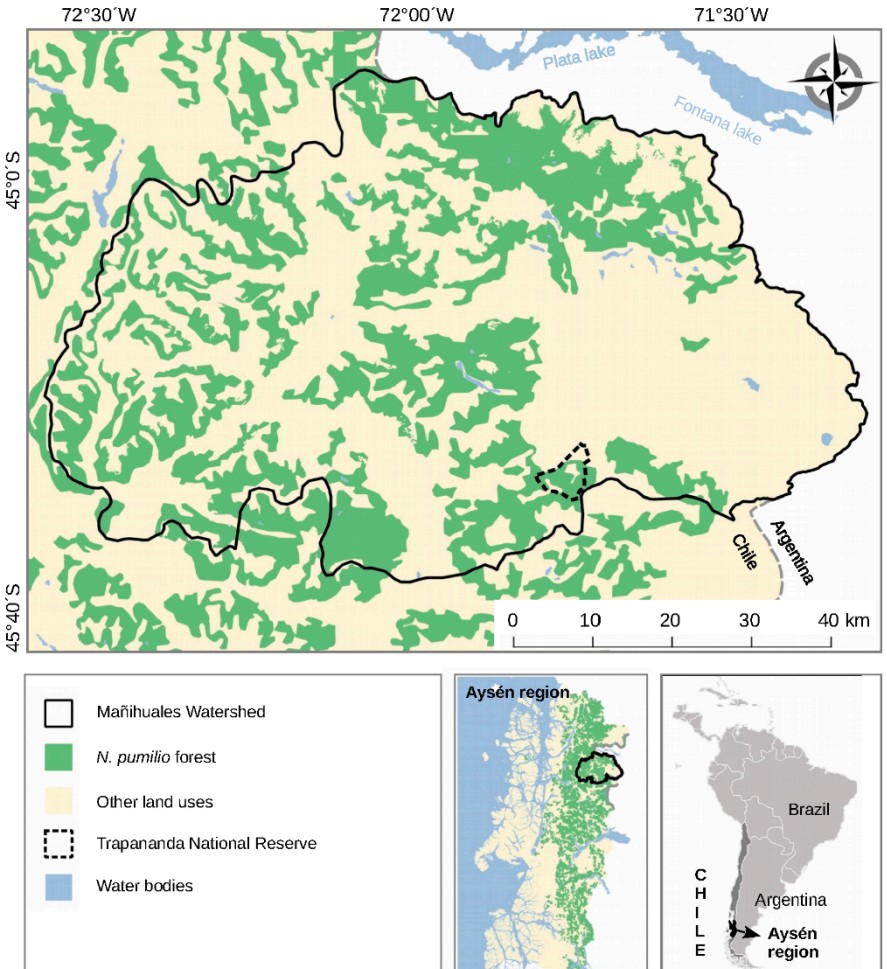

**Figure 2.** Study area corresponding to the Mañihuales watershed in the Aysén Region, Chilean Patagonia. The Trapananda National Reserve, the area where the validation campaign took place, is demarcated by a segmented black line. Land cover source: CONAF [38].

### 2.2. MODIS Time Series Data and Quality Assessment

In this study, we used the EVI as provided by the VI product of the MODIS, available at the USGS Earth Explorer geoportal. The EVI is sensitive to vegetation green biomass [39–41], and time series of the EVI provides a good proxy of the vegetative phenological cycle of broad-leaved plants [26]. On this basis, insect outbreak detection can be achieved by relating forest defoliation to negative anomalies of the EVI signal [22,25,26,42].

Several studies have shown the advantages of using high temporal resolution data from the MODIS to detect and quantify forest insect defoliation [20,26,28,39,40]. MODIS, on board the Terra and Aqua satellites (launched in 1999 and 2002, respectively), provides 16-day composites of EVI at moderate spatial resolution (250 m pixel resolution) [39]. The 16-day composites (MOD13Q1 from Terra and MYD13Q1 from Aqua) are especially useful for cloudy areas, such as Chilean Patagonia, since they are constructed with all cloud-free pixels available for each 16-day time frame, and therefore provide consistent and much more complete scenes: 23 times per year. Since Version 6, the Aqua MYD13Q1 VI composites have a time shift of 8 days to the Terra MOD13Q1 composites, and these two combined provide temporal resolutions of 8-day data worldwide. These extraordinary attributes make MODIS products an excellent alternative for near-real-time monitoring of insect forest defoliation over large areas and at a different temporal and spatial scales [20,22,27,43].

We downloaded all MODIS EVI 16-day composites available for our study area (400 scenes of the MOD13Q1 product and 346 of the MYD13Q1 product), spanning the 2000–2017 period. We used the

MOD13Q1 and MYD13Q1 combined, providing 250 × 250 m spatial resolution and 8-day temporal resolution for the growing season 2015–2016 (46 scenes), when the *O. amphimone* outbreak took place, and for most of the reference period (2002–2003 GS to 2014–2015 GS). Using the VI detailed QA band of this product and the "raster" package [44] from the R software [45], we filtered out all unuseful pixels. To achieve this, we transformed the 16-bit unsigned integer values of each pixel for each scene into binary (0–1) 16-bit code (see [46] for more details on the MODIS QA bands). We deleted all pixels with the following combinations: from bits 0–1 (VI Quality) combinations 10 and 11 corresponding to probable clouds and non-produced pixels; from bits 2–5 (VI Usefulness) combinations 1100, 1101, 1110, 1111 corresponding to the lowest qualities; from bits 6–7 (aerosol quantity) only 11 (high); from bit 8 (adjacent cloud detected) combination 1 (yes); from bit 9 (Atmosphere BRDF correction) we did not delete anything since BRDF correction is not implemented for this part of the world; from bit 10 (mixed clouds) combination 1 (yes); from bits 11–13 we deleted all combinations except 001 (land); from bit 14 (possible snow/ice) combination 1 (yes); and finally from bit 15 (possible shadow) combination 1 (yes). After this procedure, 54% of all available pixels of *N. pumilio* forests (considering all 746 scenes) were deleted, especially during winter time. From the main GS (active vegetative period), only 19% of available pixels were deleted.

### 2.3. Insect Outbreak Detection and Mapping

We developed an algorithm in R to reconstruct the annual EVI phenological cycle of *N. pumilio* forests using MODIS EVI time series for the period 2001–2015 (before the outbreak) and to calculate EVI anomalies for the 2015–2016 GS (during the outbreak) (Figure 3). This algorithm was implemented in the "npphen" R-package [47] and is illustrated in Figure 3. Details of the "npphen" package, functions and capabilities can be found in [48]. The R-package includes multi-core capabilities and handles large raster datasets. The analysis is performed at the pixel level, handling each pixel's time series individually. The procedure starts by dividing the time series into the reference period and the monitoring period (Figure 3a, red box). Then, all GSs together are arranged in the so-called EVI–day-of-the-growing-season (DGS) space and a kernel density estimation is calculated based on all available observations (Figure 3b). DGS are calculated internally by the "npphen" functions in a similar way as the "day of the year" in the Northern Hemisphere, but instead of starting on 1 January, DGS starts on 1 July. After normalization, i.e., densities from the kernel are standardized to sum 1 in the empirical cumulative distribution, this space is transformed into a two-dimensional probability distribution which represents the probability that an observed EVI value at a given DGS does not belong to the "typical" phenological cycle of that pixel. From this space, the expected EVI value per DGS (dark red line) is set as the baseline annual phenological cycle. Then, EVI anomalies can be calculated as the difference between values observed in the monitoring period and the expected annual phenology (Figure 3, black arrow). Then, the position of the observed EVI anomaly (circled point) within the kernel is used to assign its anomaly probability. In simple terms, this value can be interpreted as the probability of an observed EVI value being a "true anomaly"; for example, an "anomaly probability" of 1 means that the observed EVI value certainly does not belong to the typical phenological cycle. In Figure 3, the EVI observation circled in green falls outside the 0.95 band of confidence, which can be interpreted as a 95% confidence that the circled EVI value does not belong to the reference phenological cycle of that pixel. EVI anomalies can be transformed to a proxy of percentage of defoliation by means of percentage of EVI loss (Figure 3c). To achieve this, the minimum value of the reconstructed phenological cycle is set as the "winter" line or the "total defoliation" baseline, which corresponds to the EVI value at the time when the *N. pumilio* forests have no leaves. The R package implementation of our method utilizes EVI time series raster stacks as input data and delivers EVI anomaly raster stacks of 46 layers per GS (each layer corresponds to a DGS) along with anomaly probability raster stacks also with 46 layers per GS.

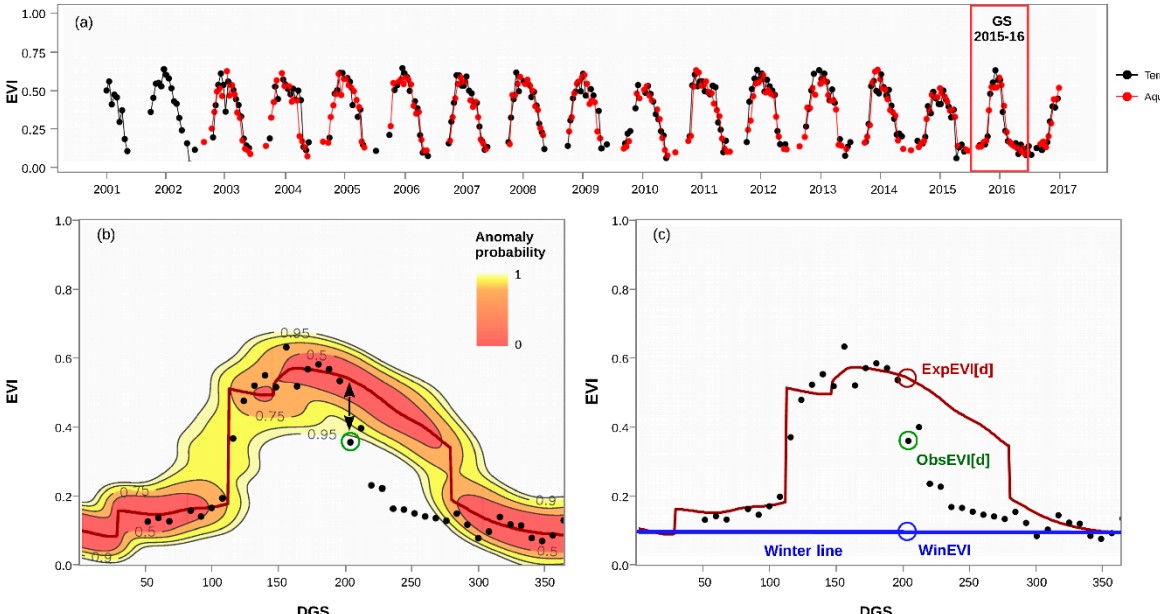

**Figure 3.** Calculation of EVI loss percentage using a probabilistic non-parametric approach and MODIS time series. (**a**) A dense time series of MODIS EVI Terra and Aqua combined (8-day time step) of a broad-leaved *Nothofagus pumilio* forest. (**b**) Kernel density estimation (time-EVI space) showing expected EVI values at different probabilities; the most probable EVI value or expected EVI per DGS corresponds to the red line; black dots correspond to the 2015–2016 GS when the outbreak took place. (**c**) Self-calibrated EVI loss calculation is achieved by calculating the Winter EVI as the minimum ExpEVI of the growing season (blue line), the EVI growth per time step, (ExpEVI[d] − WinEVI), and the observed EVI anomaly (ExpEVI[d] − ObsEVI[d]). Finally, the % of EVI loss is calculated as (ExpEVI[d] − ObsEVI[d])/(ExpEVI[d] − WinEVI).

### 2.4. Field Validation and Field Leaf Area Index Data

CONAF carried out a field campaign at the beginning of February 2016 to capture GPS points at locations where the *O. amphimone* outbreak caused total forest defoliation. From the local forest rangers, we learned that this insect outbreak event massively defoliated the forest, leaving no partially defoliated forest behind. The GPS points were used as present/absent data to visually check the spatial agreement between the anomaly map and ground observations.

Additionally, 21 field measurements of Leaf Area Index (LAI) for different *N. pumilio* stands were conducted during 2015–2016 to fit an empirical equation between LAI and MODIS EVI measurements (Figure 4). The measurements were carried out using a LaiPen instrument (Photon System Instruments, Czech Republic). Each LAI measurement consisted of 75-point single LAI measures distributed systematically in three 50-meter transects with 20 meters between transects. This design was set to capture a representative LAI measurement for the dimensions of a single MODIS pixel. Finally, the empirical curve shown in Figure 4 was used to estimate LAI anomalies from EVI anomalies and this way provide maps and statistics of defoliation in terms of LAI loss (%).

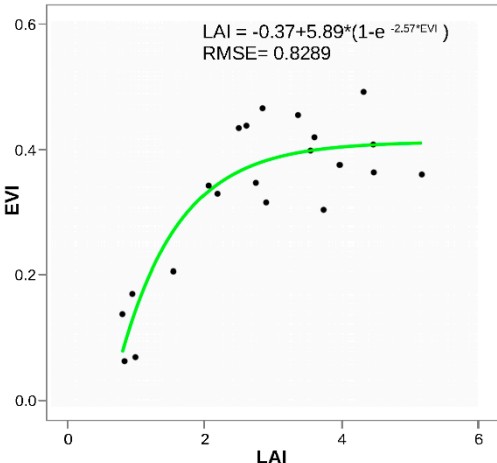

**Figure 4.** Empirical relationship between the Enhanced Vegetation Index (EVI) and in situ measurements of leaf area index (LAI) of *N. pumilio* stands.

## 3. Results

### 3.1. Calculation of EVI Loss (%) and Anomaly Probability

For each pixel of the Mañihuales watershed, we calculated the EVI loss (%) at 46 dates during the 2015–2016 growing season, when the *O. amphimone* outbreak occurred. Along with the EVI loss (%), we calculated its quality assessment band or "anomaly probability" band, which provides the position of the EVI observation within the historical EVI frequency distribution at that date (Figure 3). The critical "anomaly probability" for which an EVI loss (%) can be judged as "true" or "significant" is defined by the user. In the present study, we set this value at 0.9, a relatively high value, in order to maximize specificity and thus minimize commission error. Consequently, EVI loss (%) values with "anomaly probability" values < 0.9 were filtered out (Figure 5). Similar to other threshold selection procedures, in this case there is no general criterion to define a given value. It depends on the study objectives. In our case, the value of 0.9 was set considering the practical issue of minimizing the number of false positives in an area as remote as Aysén, where field validation is very difficult to achieve. However, for other purposes, maximization of sensitivity could be the right choice (e.g., for early detection of invasive species), in which case a lower value could be defined (e.g., 0.75).

An example of the use of the quality assessment band is given in Figure 5. At the end of the summer (DGS 268, 21 March 2016), the defoliated area of *N. pumilio* forest in the Mañihuales watershed reached its maximum extent and intensity (Figure 5). Figure 5a shows the EVI loss (%) before the anomaly probability flag is used to mask out "non-significant" values. Figure 5b shows the associated anomaly probabilities and Figure 5c the "clean" EVI loss (%) where values with "anomaly probability" <0.9 were filtered out. Finally, Figure 5d shows the LAI loss (%) after EVI values were transformed using the empirical equation given in Figure 4. When applying the anomaly probability band at this date (DGS 268), it was mostly pixels at the lowest LAI or EVI loss (%) categories that were filtered out. In fact, for the 1–20%, 20–40%, and 40–60% LAI loss categories, 99%, 96%, and 74% of the pixels were filtered out, respectively, whereas for the 60–80% and >80% LAI loss categories, only 36 and 24% of the pixels were filtered out, respectively. Another relevant aspect to report about the application of the anomaly probability band is that more pixels are filtered out as the summer moves towards the autumn (Figure 6). This can be explained by natural variations in the timing of the growing season's offset along the reference period. Thus, more pixels at the 2015–2016 GS present EVI loss (%) values, falling within the normal range towards the end of the summer.

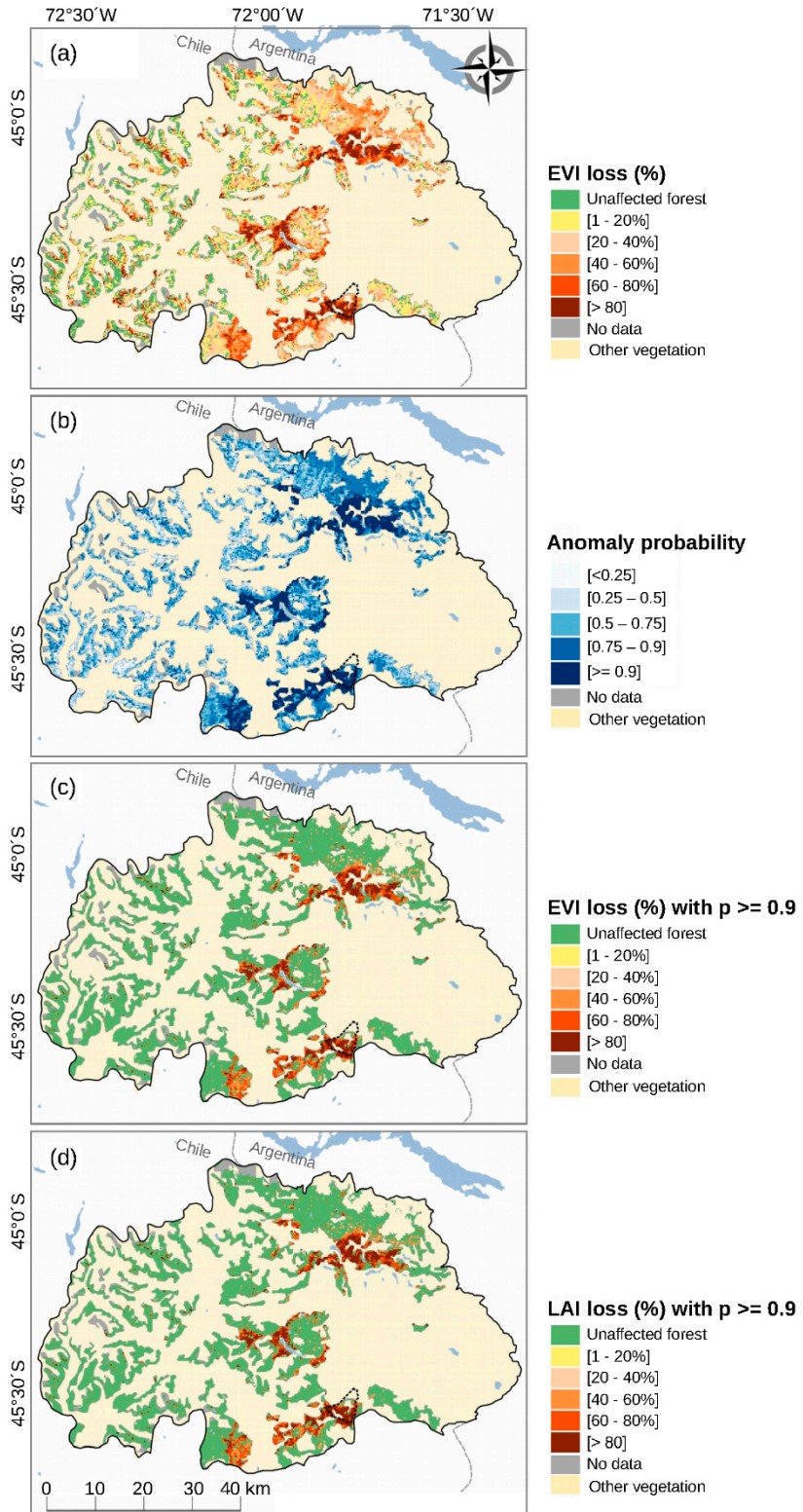

**Figure 5.** Maps displaying the level of forest defoliation caused by *O. amphimone* caterpillars at the end of the summer (DGS 268, 21 March 2016). (**a**) EVI loss (%) before the anomaly probability band is used to mask out "non-significant" values; (**b**) anomaly probability map; (**c**) EVI loss (%) with anomaly probability ≥ 0.9; (**d**) LAI loss with anomaly probability ≥ 0.9.

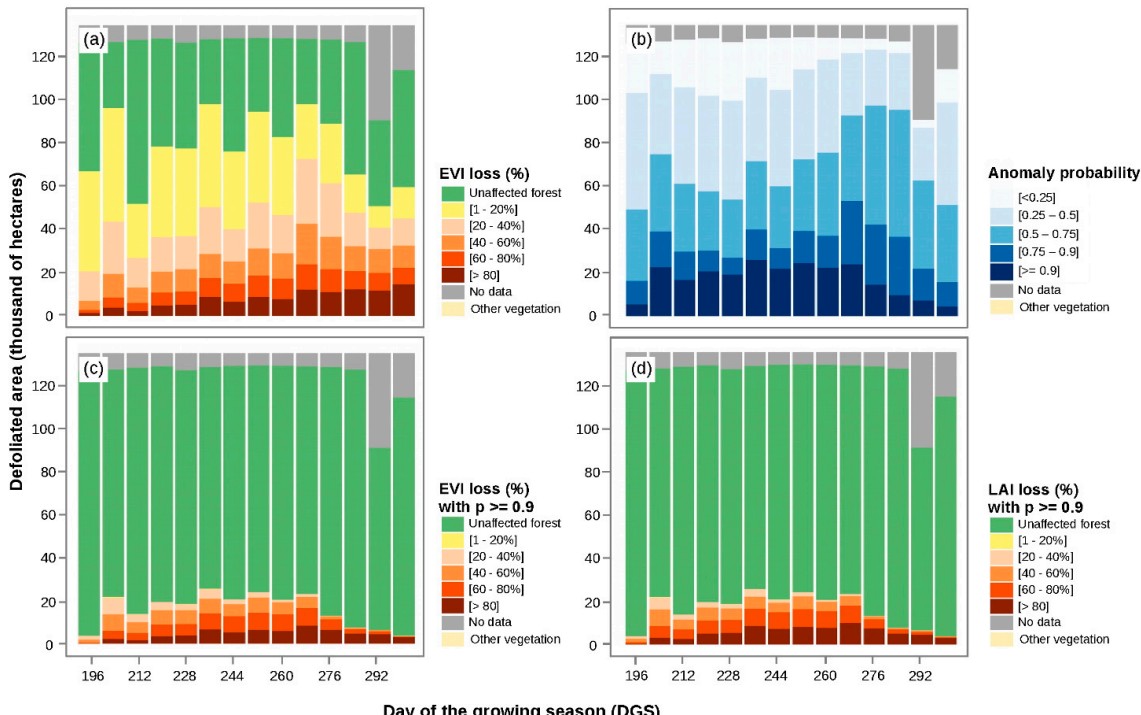

**Figure 6.** Surface of *N. pumilio* forest affected by different levels of *O. amphimone* defoliation during the summer 2015–2016 (DGS 196: 9 January 2016 to DGS 292: 15 April 2016). (**a**) EVI loss (%) before the anomaly probability band is used to mask out "non-significant" values; (**b**) anomaly probability values (in the present study only EVI loss (%) values with probabilities $\geq$ 0.9 of being "true" defoliation were considered); (**c**) EVI loss (%) with anomaly probability $\geq$ 0.9; (**d**) LAI loss with anomaly probability $\geq$ 0.9.

### 3.2. Spatiotemporal Patterns of the O. amphimone Outbreak

Figures 6 and 7 show the spatiotemporal patterns of the *O. amphimone* outbreak during the summer 2015–2016. We identified four heavily affected sectors: one in the north-east part of the Mañihuales watershed (close to the border with Argentina), one in the center, and two in the south. Of these last two, one corresponded to the Trapananda National Reserve (Figure 7, dotted area). The outbreak started in DGS 196 (9 January) in two sectors of the watershed (Figure 7, center and north) and spread rapidly to the other three sectors. In only 40 days (between DGS 196 and 236), *O. amphimone* caterpillars defoliated 15,387 ha of *N. pumilio* forest (considering only the highest LAI loss categories, 60–80% and >80%) with an average rate of 384 ha/day. Considering the same two highest LAI loss (%) categories, the outbreak reached its maximum area (17,850 ha) at the end of the summer (DGS 268, 22 March).

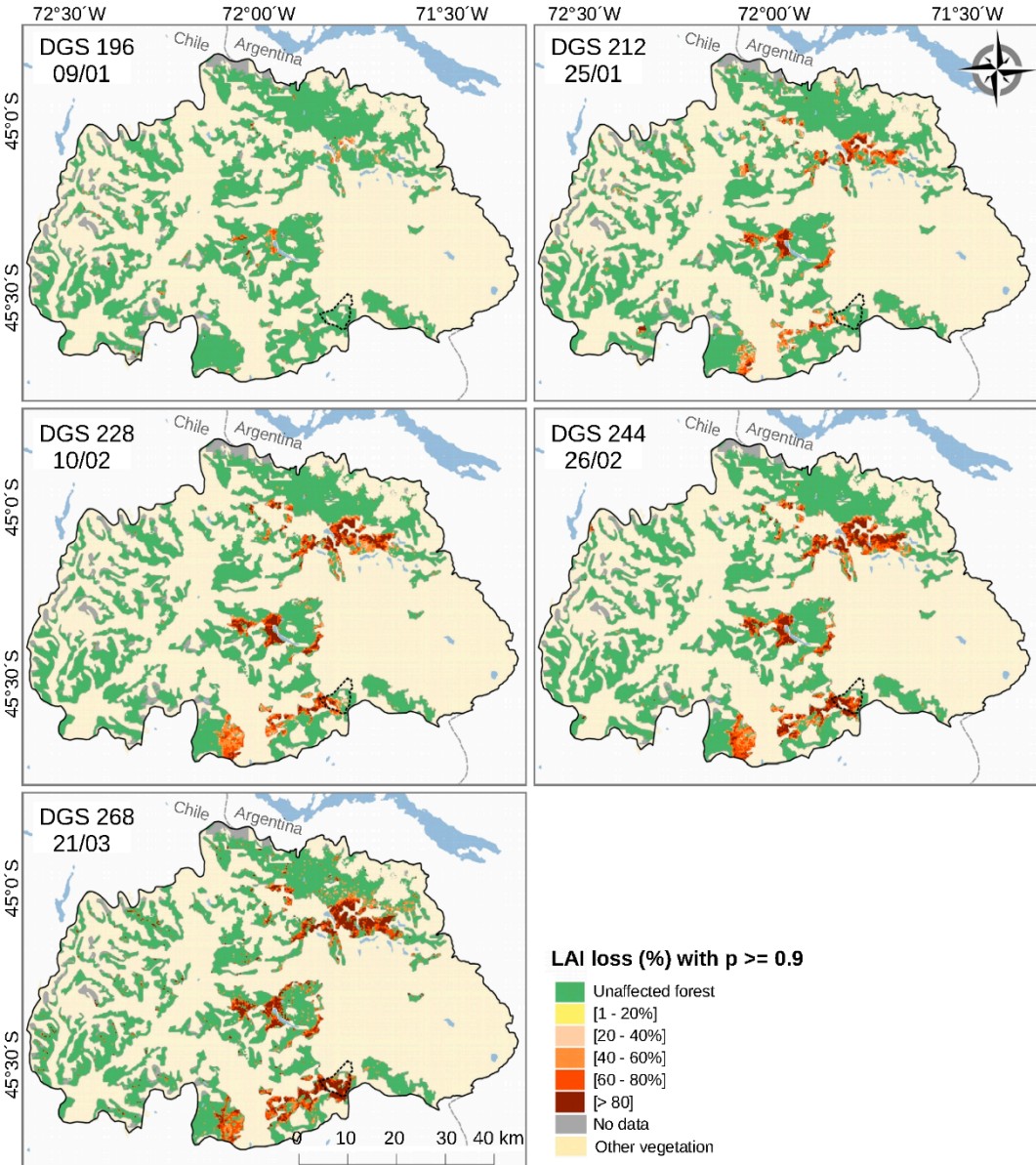

**Figure 7.** Spread of the insect outbreak during the 2015–2016 growing season in terms of LAI loss (%).

*3.3. Remote Sensing and Field Measurements of Defoliation in the Trapananda National Reserve*

During the last days of January 2016, forest rangers from the Trapananda National Reserve reported an aggressive eruption of *O. amphimone* caterpillars in native *N. pumilio* forests and we quickly organized a field campaign to record points where total defoliation was visually observed. The position of these points is shown in Figure 8 together with LAI loss (%) corresponding to the 8-day composite (DGS 228) closest to the date of the field campaign. Visually, we considered a fairly good spatial agreement between the field observations and the remote sensing assessment considering that for some points, GPS positions were taken from a safe distance to the forest due to the harsh field conditions, i.e., millions of caterpillars on the floor, in trees and even falling from the trees' crowns (Figure 1). *O. amphimone* caterpillars have urticating hairs and they can cause severe allergic reactions.

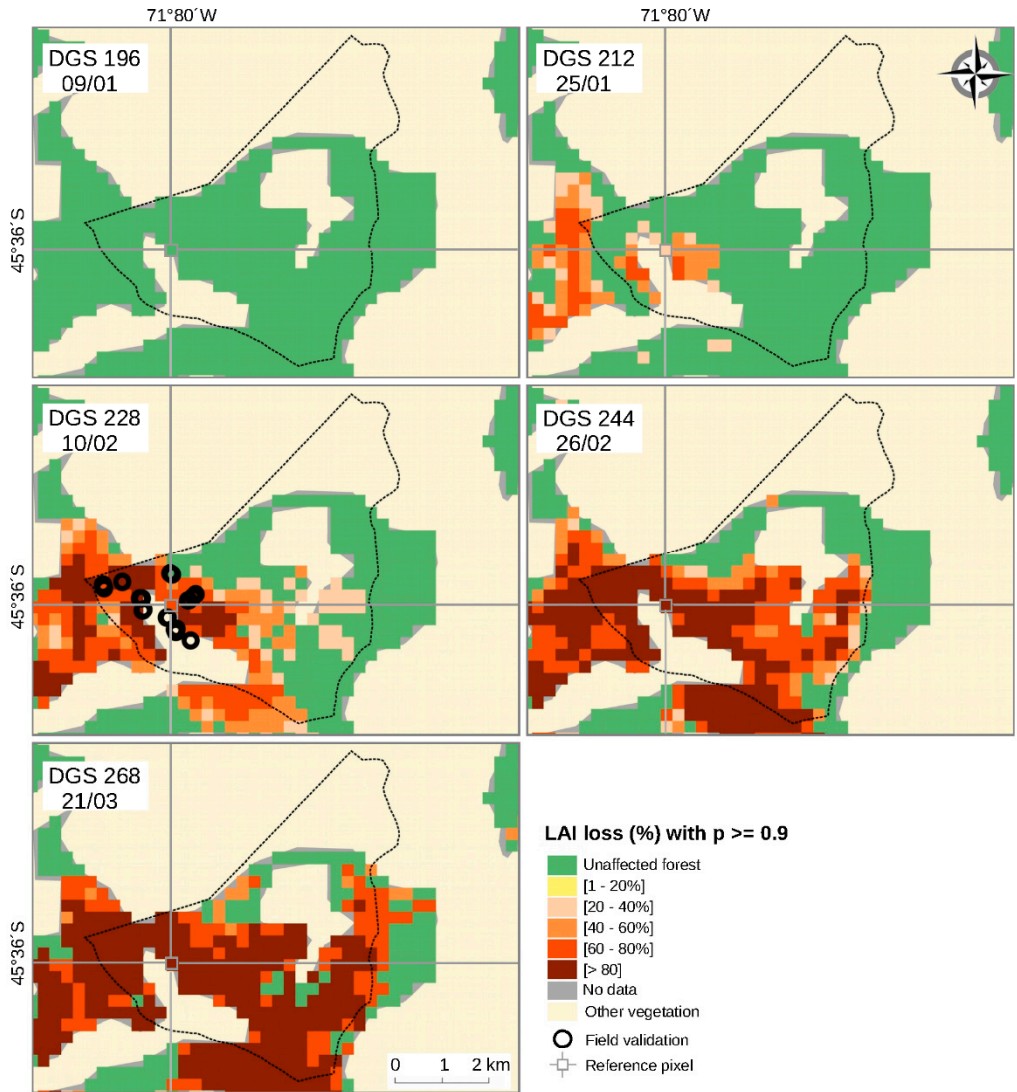

**Figure 8.** Spread of the insect outbreak during the growing season 2015–2016 within the Trapananda National Reserve. The reference pixel is the location of the time series shown in Figure 3. Black circles show the location of field observation of total defoliation caused by *O. amphimone* caterpillars.

The remote sensing assessment showed that the *O. amphimone* outbreak reached the Trapananda National Reserve from the east on DGS 204 (17 January) and spread eastwards (Figure 8), reaching a maximum defoliated area of 1050 ha on DGS 268 (22 March). This area corresponds to 77% of the total surface of *N. pumilio* in the reserve and only considering the highest defoliation ranges (60–80% and >80%). The field observations of total defoliation carried out on DGS 228 are consistent with the pest advance line at that date.

## 4. Discussion

### 4.1. Performance of the Self-Calibrated Non-Parametric Approach

The method presented in this paper allowed us to successfully detect and map the massive *O. amphimone* outbreak in the Trapananda National Reserve and three other sectors of the Mañihuales watershed during the austral 2015–2016 GS. By using dense time series of MODIS 16-day composites from the Terra and Aqua satellites combined, we constructed maps at 8-day temporal resolution, evidencing the spread of the outbreak at a temporal resolution suitable for near-real-time detection. Similar to other methods used to detect insect defoliation using dense vegetation index time series

analyses [25–27], our approach is based on detecting negative anomalies from a given annual phenological baseline, given by a drop in the VI (EVI in our case) due to defoliation. However, there are two aspects of our method that make it an advantageous alternative: First, it does not use parametric models, such as cosine [22,27], double logistic [26], or harmonic [25] functions, to set the annual phenological baseline, but instead a probabilistic approach which delivers the most "expected" annual phenological baseline together with its confidence interval, adjusting to all kinds of annual phenological behaviors. It has been pointed out that most of these parametric approaches have been developed for temperate forests with bell-shaped annual phenological curves, which may not be suitable, for example, for arid or semi-arid vegetation [31]. Second, the confidence interval around the most expected annual phenological behavior provided in our method enables the calculation of the anomaly probability of the observed VI anomaly. These probabilistic band values (ranging from 0 to 1) provide critical information to judge the anomalies as "true" defoliation (high anomaly probabilities) or as part of the natural variability of the phenological baseline (low anomaly probabilities). Furthermore, the critical anomaly probability value to judge the anomaly as "true" or "false" can be set by the user. Some users may be interested in finding "complete" anomalies (values never reported before for a given moment of the growing season), i.e., anomaly probabilities of 1, while other may consider strict values to flag "true" anomalies, e.g., anomaly probabilities of 0.95 or 0.9 as in our case. Parametric approaches calculate only the anomalies, but they do not deliver a reliability band along with the anomalies reported.

To give an idea of the impact of using different values of our anomaly probability band on the output defoliation map, we calculated the defoliated area at different levels of defoliation when using "anomaly probability" values ranging from 0.75 to 1 (Figure 9) for DGS 268. Especially for the lower defoliation categories, the differences in the reported defoliated area changed dramatically. Even considering only the highest defoliation categories (60–80% and >80%), the reported defoliated area changed from 24,068 ha when no anomaly probability band is applied to 22,681 ha for $p < 0.75$ and to 17,850 ha for $p < 0.9$ (the value used in this study). Since studies using parametric approaches do not provide an "anomaly reliability" band, it is difficult to filter out unreliable anomalies, and consequently the reported defoliated areas are likely to be overestimated.

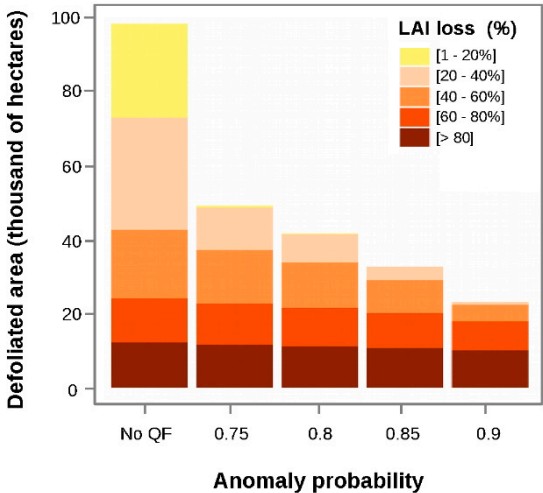

**Figure 9.** Defoliated area on day of the growing season (DGS) 268 when applying different values of the anomaly probability flag.

## 4.2. Considerations about the VI Time-Series Quality

Our approach needs several growing seasons (>3) to calculate the probabilistic kernel and the annual phenological baseline. For this reason, longer time series are more suitable for our approach, providing more robust results. For short time series, parametric approaches are more advantageous since parametric curves can be fitted even to VIs data from a single growing season if the user deems it

to be representative of the annual phenological baseline. Another relevant aspect to consider when applying our approach is that the input time series should be as "clean" as possible, i.e., without data contaminated by atmospheric effects or signal saturation, or with pixels corresponding to clouds, cloud shadows, water, ice or other elements different from the target forest. All noisy data will have an impact on the confidence intervals of the annual phenological cycle. It is therefore highly recommended to use the quality assessment bands provided with the time series products (e.g., the VI detailed QA band of the MODIS Vegetation Indices product used in this study) to filter out undesirable data. The phenological reconstruction implemented in our approach handles missing data and there is no need to complete or smooth incomplete time series after being cleaned using their quality assessment bands. Furthermore, a few outliers will not have a significant impact on the baseline phenological curve, especially for long and dense time series, but when present in the monitoring period they can flag false negative insect defoliation on the output maps and therefore, they should be handled carefully (e.g., by evaluating the next DGS to confirm the anomaly).

Figure 10 shows the EVI time series and the reconstructed annual phenological baseline for a "dirty" time series where no MODIS QA is applied (Figure 10a–d), a "clean" EVI time series with only "good data" as flagged by the MODIS pixel reliability band (Figure 10b–e) and another "clean" EVI time series, but this time using the MODIS VI detailed QA and the criteria explained in Section 2.2 (Figure 10c–f). The MODIS pixel reliability band has only 4 possible values (of which 0 corresponds to "good data") and it can be directly used to filter out unuseful pixels whereas the VI detailed QA band requires a transformation of the integer values to their binary code before it can be used. From a visual assessment of the kernel estimations, one cannot see dramatic differences on the annual phenological baseline except for the winter period. When applying the MODIS pixel reliability band to clean the EVI time series, hardly any observation remained in the winter period and this is the reason why we used the VI detailed QA to "save" some winter records by including pixels with medium and low aerosol levels. This way, and besides the VI detailed QA is more difficult to apply than the pixel reliability band, we aimed at describing better the winter part of the baseline curve [46]. Overall and at least for this example, all options allowed us to detect and assess the insect outbreak, showing the robustness of our method.

The capability of the MODIS QA bands to detect unuseful pixels will certainly have an impact on the reconstructed annual phenological baseline and its confidence intervals (see Figure 10). However, the presence of outliers (Figure 10a) will have a lower influence on the calculation of the annual phenological baseline compared to parametric approaches. In parametric methods, every point is equally weighted for the parameter estimation (unless the data analyst specifies something different). Therefore, the expected value is always conditioned in some level by outliers and especially for their magnitude (e.g., how far is from the historical mean). From here arises the common recommendation of removing outliers previous to model fitting in parametric methods. In the kernel estimation, and in non-parametric methods in general, outlier magnitude has a lower effect on the expected baseline because they depend on density. Therefore, outliers will have an impact only if they are highly frequent. A good review about the influence of outliers on parametric and non-parametric test can be found in [49]. Nevertheless, more research needs to be done to compare our approach to parametric methods at different forest sites and climate conditions.

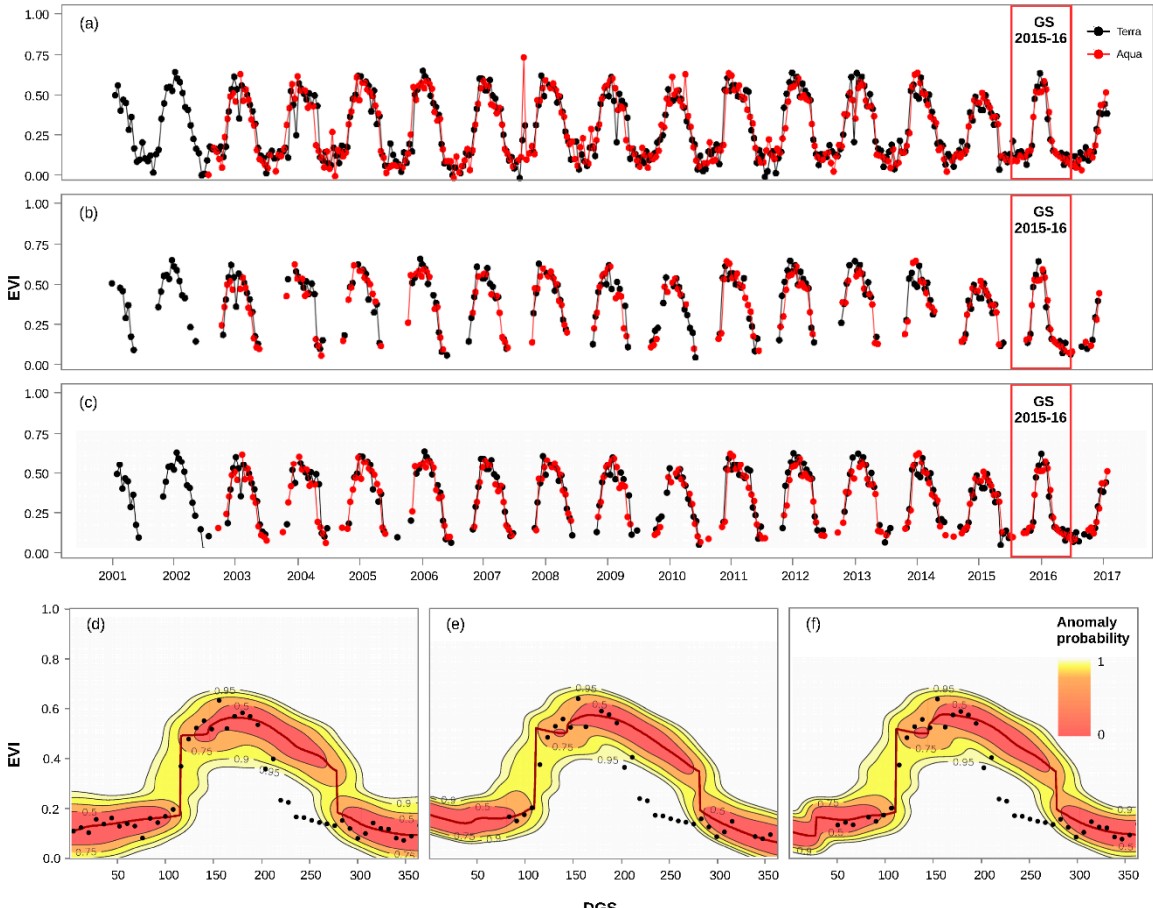

**Figure 10.** EVI time series (**a–c**) and kernel estimation (**d–f**) using different schemes of MODIS quality assessment (QA) bands: (**a–d**) with no QA applied, (**b–e**) with QA using the pixel reliability band (only "good data" was considered), (**c–f**) with QA using the VI detailed QA and the criteria explained in Section 2.2 (pixels with low and medium aerosol contamination were also considered).

### 4.3. Performance of EVI for Detecting LAI Loss Due to Defoliation

A secondary contribution of our work is to provide insights into the relationship between EVI and in situ measurements of LAI for *N. pumilio* forests. As shown in Figure 4, there is a saturation of EVI at LAI values higher than 3 which is in line with other studies showing this saturation effect [50–53]. However, both EVI anomaly loss (%) and LAI loss (%) provide similar maps regarding intensity and spatial distribution of the *O. amphimone* outbreak (Figure 5c,d), being equally helpful for forest health monitoring. The baseline annual curve in terms of EVI (a) and LAI (b) are displayed in Figure 11, where the EVI curve shows a sharper slope between DGS 180 and 280 than the LAI curve due to the saturation effect for LAI > 3. Apart from this main difference, the two curves look very much alike and anomaly detection in terms of "probability anomaly" occurred at the same time, although with differences in the loss (%), particularly between DGS 180 and 280. Overall EVI loss (%) is a good indicator of LAI loss (%) for *N. pumilio* forests and could be used as a proxy for defoliation in other studies of this widely distributed species.

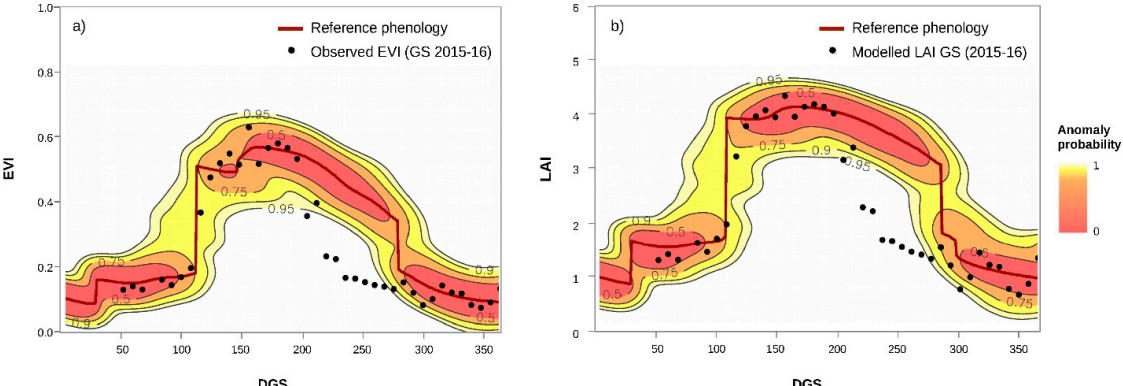

**Figure 11.** EVI (**a**) and LAI (**b**) reconstructed baseline phenology and observations during the *O. amphimone* outbreak for a single MODIS pixel.

### 4.4. Opportunities for Forest Pest Management

For remote and isolated areas like Chilean Patagonia, with human density as low as <1 inhabitants/km$^2$ [54], semi-automated tools for forest health monitoring represent a viable alternative to field surveys. Contrasting with the usually expensive and time-consuming field surveys, remote sensing-based methods like the one introduced here could imply a significant cost reduction in monitoring efforts [55]; therefore, budget could be reassigned to preventive actions. In our case study, the Aysén Region is one of the most isolated areas in South America, with a forested area as large as the entire country of Portugal or Ireland. For such vast and remote areas, the use of remote sensing methods is crucial for effective and robust outbreak detection and quantification. As we present here, the combination of robust statistical methods with dense time series such as MODIS EVI or NDVI enable insect outbreak detection on a near-real-time basis (e.g., 8-day time window). The flexibility of the method to handle time series with and without gaps, makes it a good alternative for applications at higher spatial resolution: for example the Landsat dataset provide temporally irregular time series, depending on cloud cover, for the period 1984-present at 30 m pixel resolution [56] and the Sentinel-2a-b at 10 meters pixel resolution [57] for the period 2015–present for 2-a and 2017-present for 2-b. The Sentinel 2-a-b dataset has great potential for insect outbreak monitoring using our approach considering its revisit time of 5 days, but it will need more years to provide enough GSs for the kernel estimation. The method presented here enables detection of both outbreak spatial distribution and species geographical range expansion, two key variables for understanding insect outbreak dynamics [58]. Concerns about the impact of climate change on outbreak dynamics and the urgent need for early detection of these changes make the implementation of near-real-time monitoring via remote sensing a real alternative for local and national pest management as well as for issues related to forest health.

### 4.5. Potential for Future Forest Insect Outbreak Research

Although applicable to all kinds of broad-leaved and evergreen forests, our approach is presented here as an alternative for evaluating insect defoliation of deciduous broad-leaved forests with no reference field data. The strict deciduous habit of *N. pumilio* provided a unique opportunity to use the EVI values recorded during the winter recession for setting the leafless condition on a pixel basis. This way, the EVI loss (%) and, after applying an empirical equation, LAI loss (%) are finally calculated. The "self-calibrated" nature of our approach makes it a powerful tool for monitoring deciduous broad-leaved forests in remote and inaccessible areas like Chilean and Argentinian Patagonia with little or no field data on the outbreak events. Although EVI anomalies and anomaly probability bands can be calculated for all type of forests, field data will be required to translate the EVI loss into a defoliation percentage.

Our non-parametric approach can be particularly useful to further study the apparently increasing frequency of outbreaks due to warming trends in Patagonian forests [59], opening new opportunities for long-term monitoring of climate change impacts in these remote forests. Future research in the same region can be conducted using the method presented here to reconstruct past outbreaks (e.g., [60]), or to conduct a near-real-time assessment of an ongoing insect outbreak in the future (e.g., [25]), or expanding our results to a regional scale (>100 km$^2$) in deciduous broad-leaved *Nothofagus*-dominated forests in the Mediterranean region of South America [34], where recent insect outbreaks have also been detected but not yet studied as detailed as the present contribution. Such studies can help to disentangle climate-driven from biotic-driven regional-scale disturbances in the temperate forests of South America, where deciduous *Nothofagus* tree species are dominant and have comparatively high disturbances compared to other temperate forest landscapes [61].

## 5. Conclusions

We introduced and tested a new non-parametric time series method to detect and map insect forest defoliation using time series of satellite VIs. Unlike commonly used parametric approaches, our method provides a robust assessment of the defoliation level by means of VI anomalies from the undisturbed regular annual phenology along with an anomaly probability output band indicating the probability that the observed VI anomalies fall outside the range of forest phenological natural variability. This anomaly probability band is based on the frequency distribution of all historical VI observations and can be used to neglect VI anomalies at different significance levels according to users' needs. The method handles all kinds of time series, with regular or irregular time steps, with complete series or with missing values, taking full advantage of the temporal richness of currently available satellite products. Its flexibility in describing different annual phenological behaviors (no need for pre-defined parametric curves) and the possibility of self-calibrating the defoliation level in the case of deciduous forests (by using the winter VI values as the leafless reference) make our approach a valuable alternative to assess and monitor defoliation of large and inaccessible forests worldwide and can be used for implementing near-real-time monitoring systems.

**Author Contributions:** Conceptualization, R.O.C. and S.A.E.; Methodology, R.O.C., R.R. and S.A.E.; Software, R.O.C. and S.A.E.; Validation, R.R. and M.D.; Formal Analysis, R.O.C. and R.R.; Investigation, R.O.C., R.R., S.A.E., A.G.G. and M.D.; Resources, R.O.C., S.A.E. and M.D.; Data Curation, R.O.C., R.R., S.A.E., A.G.G. and M.D.; Writing-Original Draft Preparation, R.O.C.; Writing-Review & Editing, R.O.C., S.A.E. and A.G.G.; Visualization, R.O.C. and R.R.; Supervision, R.O.C. and S.A.E.; Project Administration and Funding Acquisition, R.O.C. and S.A.E.

**Funding:** R.O.C. was funded by CONICYT PAI N°82140001 (convocatoria 2014) and FONDECYT iniciación N°11171046; S.A.E. was funded by FONDECYT regular N°1160370 and CAPES-Conicyt FB-0002 (line 4) and A.G.G. was funded by FONDECYT regular N°11150835.

**Acknowledgments:** The authors wish to thank Helen Lowry for revising the manuscript and three anonymous reviewers for the thorough revision of the manuscript.

**Conflicts of Interest:** The authors declare no conflict of interest.

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
