# Peer review of "A Self-Calibrated Non-Parametric Time Series Analysis Approach for Assessing Insect Defoliation of Broad-Leaved Deciduous Nothofagus pumilio Forests"

_remotesensing, doi:10.3390/rs11020204_

Reviewer 1 Report

Dear authors,

thank you for a nice and interesting paper.

Authors presented a non-parameteric method for analysis of high density time series data of MODIS EVI index to detect vast forest defoliation in Chile.

Paper is clearly written. Methods are explained and I found just a few issues that shall be better clarified (please, see the attached pdf with my comments written directly in your text. Results are clearly presented, figures support well the text.

Although I don't consider myself as being expert in time series analysis of RS data, I think it is a nice example of how dense time series can be used for ecological monitoring, especially in such a remote areas like Patagonia.

The paper fits well to the overal scope of the MDPI RS special issue. I recommend this paper for publishing after the minor changes will be addressed. 

Author Response

Thank you for your nice and constructive comments. In the attached pdf, we addressed point by point all questions and comments. With best regards, Roberto O. Chávez

Reviewer 2 Report

The authors present a method to monitor insect outbreaks in deciduous forests with time-series of MODIS derived (both Terra and Aqua) Enhanced Vegetation Index (EVI) data. The method in non-parametric and utilizes EVI from pre-outbreak years to model the phenological cycle including confidence intervals for the expected values at each time-period. The method then calculates the probability that an EVI observation from a monitoring year is an outlier, i.e. in this case if there is an insect outbreak. The authors also demonstrate how the method can be applied to monitor the spatiotemporal development of an insect outbreak and studies the relationship between EVI and Leaf Area Index (LAI).

The article is generally well written and the developed method is interesting and has advantages as the authors state, but there are major limitation of the methods that must be mentioned in the article:

1. The method depends heavily on the quality assessment (QA) data from MODIS (which is briefly mentioned in the discussion). MODIS QA are not fully reliable and it is rather common that data flagged as high qualify are outliers (generally too low values) and that will influence the presented method. Did the authors study how accurate the MODIS QA flags are in the study area? That would be valuable information for a reader. A simple test would be to plot time series of the data that were considered as high quality and check if there are any outliers e.g. very low single observations with the observations before and after being high. At least the authors need to address that the reliability of MODIS QA data will/can influence the result.

2. Since the method heavily depends on the QA data a more detailed description explaining how the QA data were handled must be provided. It would also be interesting for a reader to know how much of the data that were considered good quality in the main growing season. Now it is only state that 54% of the pixels (mainly in winter) were filtered out (P6, L216-221).

3. Why were no attempts made to remove outliers from the original data after the low quality data were removed? That could remove EVI values where the QA data failed to identify clouds etc. and it could give more reliable expected value trajectory and confidence intervals. This could be applied also to remove outliers in the EVI data for the year when the defoliation is monitored if data are used in a real-time application; in a real-time application it is not easy to say if a value is due to defoliation or low quality data.

4. One advantage of parametric methods is that outliers can be handled when a function is fitted to smooth the raw data. This is not only an advantage, e.g. limitation in the shape of a seasonal trajectory as the authors state, but it makes parametric methods less sensitive to the accuracy of MODIS QA data compared to the developed method.

5. There is a dip in the expected trajectory of EVI early in the growing season (P8, Fig.3). This trajectory of the expected EVI values does not seem like a natural growing season for vegetation? For 2015-16 there seem to be no such dip; in the entire time-series it is not easy to see in the current figure. Is it possible that some (low) outliers in the MODIS data included when estimating the trajectory could create this early dip in the trajectory? Can it be an early signal from undergrowth that gives a high early growing season EVI value? This shape of the seasonal trajectory should be discussed.

6. In the current study with heavy defoliation in the later part of the growing season and no re-foliation the dip in the seasonal trajectory (#4 above) does not influence detection accuracy. However, for insect outbreaks that occur early in the growing season and when re-foliation is common (i.e. there is a short window with few satellite observations available to detect the defoliation) it could have a major impact on the ability of the method to detect defoliation if outliers are included and influence the expected values as well as confidence intervals. This is a problem also for parametric methods but it should be mentioned in the discussion.

P1, L24-27: There are studies where the natural variability of a vegetation index on pixel-level have been considered (using Z-score) for insect defoliation monitoring: Olsson P.O., Kantola T., Lyytikäinen-Saarenmaa P., Jönsson A. M., Lars Eklundh L., 2016. Development of a method for monitoring of insect induced forest defoliation – limitation of MODIS data in Fennoscandian forest landscapes. Silva Fennica vol. 50 no. 2 article id 1495.  https://doi.org/10.14214/sf.1495

P1, L31 (and later places): I do not really like the term "quality flag" to indicate the probability that an observation is an anomaly. The term Quality Assessment (QA) is used for the quality of MODIS data and sometimes the term QA flag is used to refer to the quality of the MODIS data; hence, I would prefer to use another term for the anomalies to clearly separate it from MODIS QA in this study. Maybe something referring to probability or reliability?

P6, L202: Here you mention that MOD13 Q1 included NDVI, and several studies you mention have also used NDVI. If you mention NDVI here you also need to give a motivation to why you used EVI instead of NDVI. Did you test with NDVI? Why did you chose EVI in this study?

P8, Fig3. The growing season 2014-15 seems to have lower EVI values than the 2000-2014 part of the time-series. Any possible reason for this? Something that triggered the large outbreak in 2015-16? Increasing larvae populations?

P16, L 399-402: "Since studies using parametric approaches lack of a quality flag, there is no way to filter out unreliable anomalies and consequently the reported defoliated areas are likely to be overestimated." - But when thresholds are used to classify pixels as defoliated or not, the threshold can be adjusted in a similar fashion as your "quality flags" to adjust how many pixels that are classified as defoliated so "no way" feels like a strong statement. If e.g. the QA data from MODIS is poor an EVI observation that is low due to e.g. clouds is flagged as reliable your method risk classifying it as an anomaly that is due to poor quality rather than defoliation. This is part of the quality of MODIS QA data that I mentioned above.

Minor comments:

P1, L18: "Folivorous insects cause among". - Should be are instead of cause.

P1, L19: "forests canopies". - Skip canopies.

P1, L 23: "forest natural annual phenological cycle" - forests'

P1, L 35: "dense time series of the MODIS Enhanced Vegetation Index (EVI)" - ....(EVI) from the MODIS sensor...

P2, L65: "defoliation level of such outbreaks". - ....these outbreaks....

P2, L65: "A major reason of this research gap" - ....reason for...

P2, L71-73: "Defoliation has an effect on the light reflection and absorption properties of the vegetation canopy, especially in the near-infrared region, which can be retrieved by multispectral and hyperspectral sensors" - How about LAI? Reformulate so that it is more clear that the large difference in LAI has a strong impact on the spectral properties which you show in you LAI - EVI part of the study.

P3, L82: "(16-has temporal resolution),". - day, not has.

P4, L132: "to overcome these limitations, in this study we propose " - ...these limitations, we propose...

P4, L144-163: This is partly a description of the study area and gives a bit too much detail to fit in the introduction. Consider shortening this part and move parts to the study area section in methods.

P6, Figure 2. The scale bar is not easy to read in the top map.

P6, L187-: Section 2.2 MODIS time series feels like a mixture of background and data description. Try to be more concise when stating what data you use. Also consider changing the heading. Maybe "MODIS time series data" or "MODIS time series data preparation" to get a more descriptive heading?

P7, L226: "This algorithm has been implemented" - was implemented.

P7, L242: "the probability of an observed EVI value of being a" - ...EVI value being a....

P9, L280: " Along with EVI loss (%), and also per pixel, we calculated its quality" - maybe remove "and also per pixel" to give a better flow.

P13, L335: " were total defoliation was visually observed using a GPS navigator." - ...,where total defoliation was visually observed, ....

P13, L341: " caterpillars in the floor" - on the floor

P13, L347: " corresponds to the 77%" - ...to 77%...

P15, L358: " outbreak occurred in the Trapananda" - skip occurred (or ...that occurred...)

P15, L359: " and other three sectors" - ...three other...

P15, L374: "example for arid or semi-arid vegetation [31]." - is it a motivation for your method that bell-shaped phenological curves are not suitable for arid or semi-arid areas?

P16, L421: " Furthermore, few out layers will not" - ...a few outliers...

Author Response

Thank you for the detailed revision of our manuscript and the positive words about our work. We have considered adding a separate discussion section for your main point regarding the effect of the MODIS QA assessment on the outputs of our method. In the attached file, we address all comments regarding this main point and other minor observations.

Reviewer 3 Report

Review of a manuscript by R. Chávez et al.: A self-calibrated non-parametric time series analysis approach for assessing insect defoliation of broadleaved deciduous Nothofagus pumilio forests

This manuscript describes an innovative approach to detect insect defoliation in deciduous forests in South America based on rather dense time series of medium spatial resolution MODIS data. The advantage of the method is that it calculates the deviation from an expected reference value for each pixel at each time step of an observation period. The reference is taken from past measurements, and therefore confidence intervals can be used to express the deviation of a current pixel value in a probabilistic way. Existing approaches model reference periods rather than accounting for measured variability. Hence, the presented paper can be seen as an improvement or at least as a promising option.

The manuscript is well written and a promising contribution to better monitoring techniques of insect defoliation. The provided figures are of high quality and support the message of the paper. However, in some parts the paper must be restructured, and few section must be improved.

The major concern I have is the kernel density estimation of the reference. If Figure 3 is representative of the study site, then there is hardly any observation during onset of vegetation greening an also during senescence. Since phenological parameters such as start of season, end of season, or length of season may differ considerably between years, I was wondering if the kernel density estimation could be improved by considering the annual start of season? It is not written in the manuscript how day of growing season (DGS) is calculated.

The analysis presented here is based on MODIS. After removal of contaminated data, the time series have considerable gaps. As the method is capable of dealing with gaps, Sentinel-2 or Landsat data could be used alternatively – in particular for the observation period. The authors could pick up the flexibility of their method in the conclusions or discussion section. MODIS’ spatial resolution might be insufficient for insect defoliation monitoring in other study sites and settings.

The LAI anomaly detection is not clearly described. It seems you derived per-pixel per-day LAI time series by modelling LAI based on EVI according to the relationship presented in Figure 6. If so, please provide the equation. Please elaborate on that.

To my understanding, near-real time monitoring can be done with the proposed method. But it is not the core application in the presented setting. Also, “near-real time” must be defined. And – if the study aims at near-real time application – the paper must contain some statistical measure of reliability in terms of timing of defoliation detection. A near-real time product must contain information about when an outbreak occurred where, how strong and at which probability.

How do you deal with insect outbreaks during the reference period? (How) can you exclude them for a clear reference?

Minor issues:
Line 62: rephrase
Line 82: 16-day, not “16-has”
Line 120: Please define “quality assessment bands”, also to ensure this term is not mixed up with the MODIS quality bands. Pasquarella et al. (2017), for example, take deviation standardized by RMSE as measure of severity. Therefore, it is too harsh to say others don’t provide “quality assessment bands”.
Line 138: It should be noted that the anomaly can be considered true at any level of divergence. The 95% threshold should be justified in some way.
Figure 1: Please indicate when the photos were taken and who took them. They should be provided along with a map (e.g., Fig. 2) and an indication where they were taken.
Line 144-168 + Figure 1: Should be moved to the study site and methods section, respectively. The introduction section should end with a outline of the paper.
Line 150: hectares
Line 152: rephrase this sentence
Please indicate in the introduction section what happens with the defoliated trees. Do they die?
Please provide few more details about the ecosystem. Are there pure stands of N. pumilio? How do the caterpillars spread?
Figure 2: Where is the land cover information from?
Line 190: Please cite Huete et al. (2002) as reference for EVI (A. Huete, K. Didan, T. Miura, E. P. Rodriguez, X. Gao, L. G. Ferreira. Overview of the radiometric and biophysical performance of the MODIS vegetation indices. Remote Sensing of Environment 83(2002) 195-213 doi:10.1016/S0034-4257(02)00096-2). And also provide literature for NDVI.
Line 225: GS must be defined at first mention
Line 232ff: If DGS does not correspond to day of year (DOY), it must be explained how start of season (SOS) is calculated.
Figure 3: Please provide a color legend for the kernel density in b). Does a value 0.5 mean that 50% of all measurements at this particular DGS in the reference period are within this range?
Figure 3: From an ecological perspective and in particular from a near-real time monitoring perspective it would make sense to detect the earliest anomaly – which in this example seems to be about 20 days earlier than the marked point. It would make sense to provide cumulated anomalies as additional product.
Figure 3: Why is the red line in c) not smooth – like a fitted line?
Line 268ff: Please provide more details about the Lai measurement, e.g. a figure of the sampling design.
Line 288: should be 2016?
Line 293ff: Please do not jump to figures that appear later. The EVI-LAI relationship must be explained earlier (methods section).
Line 298: What does neglect mean here? Sounds like indication of an error.
Line 307: The term quality band is connoted to the MODIS quality bands. Please make clear what you mean with this term throughout the manuscript.
Figure 5: No data values are caused by clouds?
Figure 6: Should come with an equation and must be moved to the methods section. Please explain the figure in the text.
Line 335: replace were by where
Line 341: in trees
Line 345: from the west
Line 358: that occurred
Line 387: The “quality assessment” mentioned here refers to the probabilities? As said before, some authors indicate anomalies expressed with RMSE.
Line 406ff: Longer time series are only appropriate in stable forests without structural development and without irregular disturbances such as fire, thinning or insect defoliation.
Line 421: few out layers – must be “few outliers”, right?
Line 429: add full stop before “However”
Figure 10: The left-hand figure must be a), the right one b). LAI is not measured but modelled. Therefore, I’d opt for “Modelled LAI” in the legend rather than “Observed LAI”.
Line 442: O. amphimone should be in italics
Line 445: inhabitants/km²
Line 452: For such vast…
Line 472: Chilean and Argentinian

There are some double spaces throughout the text. Please check.

Author Response

Thank you for your nice and constructive comments. We have implemented most of them as you can see in the attached file. With best regards,

Roberto O. Chávez
